# Multiplexed Detection of Pancreatic Cancer by Combining a Nanoparticle-Enabled Blood Test and Plasma Levels of Acute-Phase Proteins

**DOI:** 10.3390/cancers14194658

**Published:** 2022-09-25

**Authors:** Damiano Caputo, Alessandro Coppola, Erica Quagliarini, Riccardo Di Santo, Anna Laura Capriotti, Roberto Cammarata, Aldo Laganà, Massimiliano Papi, Luca Digiacomo, Roberto Coppola, Daniela Pozzi, Giulio Caracciolo

**Affiliations:** 1Department of Surgery, University Campus Bio-Medico di Roma, Via Alvaro del Portillo 200, 00128 Rome, Italy; 2General Surgery, Fondazione Policlinico Universitario Campus Bio-Medico, Via Alvaro del Portillo 200, 00128 Rome, Italy; 3NanoDelivery Lab, Department of Molecular Medicine, Sapienza University of Rome, Viale Regina Elena 291, 00161 Rome, Italy; 4Department of Chemistry, Sapienza University of Rome, P.le A. Moro 5, 00185 Rome, Italy; 5Dipartimento di Neuroscienze, Università Cattolica del Sacro Cuore, Largo Francesco Vito 1, 00168 Rome, Italy; 6Fondazione Policlinico Universitario A. Gemelli IRCSS, 00168 Rome, Italy

**Keywords:** pancreatic cancer, biomarkers, acute phase proteins, inflammation, nanotechnology, nanoparticles, graphene oxide, electrophoresis

## Abstract

**Simple Summary:**

In this study, a multiplexed strategy based on the combination of a nanoparticle-enabled blood test and serum levels of acute-phase proteins proved to be able to distinguish pancreatic cancer patients from healthy controls with a good and sex-dependent prediction ability. This study suggests a possible role of acute-phase proteins as pancreatic cancer biomarkers and paves the way for the development of multiplexed technologies for early cancer detection.

**Abstract:**

The development of new tools for the early detection of pancreatic ductal adenocarcinoma (PDAC) represents an area of intense research. Recently, the concept has emerged that multiplexed detection of different signatures from a single biospecimen (e.g., saliva, blood, etc.) may exhibit better diagnostic capability than single biomarkers. In this work, we develop a multiplexed strategy for detecting PDAC by combining characterization of the nanoparticle (NP)-protein corona, i.e., the protein layer that surrounds NPs upon exposure to biological fluids and circulating levels of plasma proteins belonging to the acute phase protein (APPs) family. As a first step, we developed a nanoparticle-enabled blood (NEB) test that employed 600 nm graphene oxide (GO) nanosheets and human plasma (HP) (5% vol/vol) to produce 75 personalized protein coronas (25 from healthy subjects and 50 from PDAC patients). Isolation and characterization of protein corona patterns by 1-dimensional (1D) SDS-PAGE identified significant differences in the abundance of low-molecular-weight corona proteins (20–30 kDa) between healthy subjects and PDAC patients. Coupling the outcomes of the NEB test with the circulating levels of alpha 2 globulins, we detected PDAC with a global capacity of 83.3%. Notably, a version of the multiplexed detection strategy run on sex-disaggregated data provided substantially better classification accuracy for men (93.1% vs. 77.8%). Nanoliquid chromatography tandem mass spectrometry (nano-LC MS/MS) experiments allowed to correlate PDAC with an altered enrichment of Apolipoprotein A-I, Apolipoprotein D, Complement factor D, Alpha-1-antichymotrypsin and Alpha-1-antitrypsin in the personalized protein corona. Moreover, other significant changes in the protein corona of PDAC patients were found. Overall, the developed multiplexed strategy is a valid tool for PDAC detection and paves the way for the identification of new potential PDAC biomarkers.

## 1. Introduction

Pancreatic ductal adenocarcinoma (PDAC) is a pathology with a poor prognosis and progressively increasing incidence [1]. Although many advances have been made in the therapeutic field, this disease remains difficult to cure mainly because it is biologically very aggressive and is usually detected at an advanced stage. The search for new strategies for early detection of PDAC has involved many researchers in recent years [2]. The studies conducted to date have led to the identification of a large series of biomarkers that, alone or in combination with others, have proved their efficacy in the discrimination of PDAC from other pancreatic diseases (e.g., chronic pancreatitis) and in distinguishing early PDACs from advanced ones [3,4]. Despite many studies having been conducted, only prognostic biomarkers were identified [5,6]. Furthermore, the most promising results were obtained by advances in molecular technologies such as gene sequencing, transcriptomics, glycomics, and proteomics [7,8,9,10]. Of note, proteomic techniques are among the most powerful tools for potential biomarker identification. Unfortunately, despite their reliability, these markers are far from finding a daily clinic application, as they do not meet the criteria of economy and reproducibility required by the WHO [11]. Thus, researchers have focused on identifying reliable biomarkers using easily manageable techniques [12]. Our group was among the first to report diagnostic technologies based on the characterization of the protein corona (PC), i.e., the protein later that coats nanoparticles (NPs) when they are exposed to bodily fluids [13,14]. We demonstrated that the NP-protein corona is personalized as it changes from one subject to another, and pathological conditions can leave a signature inside it [15,16]. Using nanoliquid chromatography tandem mass spectrometry (nano-LC MS/MS), we identified biomarkers for PDAC within “corona proteins” [17]. However, nano-LC MS/MS presents limitations that preclude its diffusion in laboratory diagnostics [18]. To overcome these restrictions, efforts were dedicated to developing NP-enabled blood (NEB) tests (reviewed in [19]). In the NEB test, nanomaterial such as gold NPs, liposomes, and graphene oxide (GO) nanosheets are exposed to human plasma (HP) collected from cancer patients and healthy volunteers under precise experimental conditions such as incubation time, temperature, protein concentration, etc. [20]. Corona proteins are isolated by centrifugation and characterized by 1D SDS-PAGE. The one-dimensional (1D) protein pattern is segmented into regions of molecular weight (MW), and the integral areas for each region are used as input parameters for a principal component analysis (PCA) and linear discriminant analysis (LDA). The sensibility and specificity of the NEB test for PDAC are typically higher than those of carbohydrate antigen 19-9 (CA19-9), which is the only biomarker approved by the regulatory agencies for PDAC monitoring [21]. In particular, the sensitivity and specificity of CA 19-9 reach about 79–81% and 82–90%, respectively. Furthermore, considering that the multiplexed strategy that combines outcomes of clinical biomarkers proved to be effective in the early detection of neoplastic diseases [22], in 2020, an approach was proposed that correlates the outcomes of the NEB test with levels of clinical biomarkers [23]. Recently, some authors reported an increased risk of PDAC in the presence of alteration of pre-diagnostic serum levels of C-Reactive Protein (CRP) and haptoglobin, which both belong to the globulins of the acute phase protein (APPs) family. These proteins are separated from albumin during protein electrophoresis analysis [24]. Starting from these premises, here we combined a GO-based version of the NEB test with serum APPs levels. This multiplexed strategy led to a test exhibiting an impressive classification accuracy, which reached 93% for male subjects. Compared to proteomic techniques that are usually employed for the identification of PDAC’s biomarkers, the presented test represents a faster and cheaper approach, as it exhibits a low cost of investigation, high sensitivity, specificity, and reproducibility.

## 2. Materials and Methods

### 2.1. Patients’ Enrolment and Inclusion Criteria

Cyto-histologically diagnosed and proven PDAC patients admitted to the Fondazione Policlinico Universitario Campus Bio-Medico who fulfilled the inclusion criteria have been considered eligible for the study. Healthy controls have been identified among patients admitted to the Surgery Center of the same hospital for benign surgical diseases (e.g., cholelithiasis, abdominal hernia, hemorrhoids, etc.). Patients affected by acute diseases, that by their nature could influence the APPs concentration (e.g., peritonitis, bowel obstruction, etc.), have been excluded from the analysis, such as patients who underwent urgent procedures. The inclusion criteria for both groups were age ≥ 18 years; normal renal function (creatinine < 1.5 mg/dL, blood urea nitrogen < 1.5 times the upper limit), or mildly decreased renal function (<G3 based on the Glomerular Filtration Rate score proposed by the KDIGO guidelines) for less than 3 months; no previous personal medical history of malignant tumors and chronic blood disease; absence of the previous history of chemotherapy and radiotherapy; absence of uncontrolled infections, pancreatitis, lung, and hepatic coexistent diseases. Written informed consent was obtained from all the participants. Demographic and clinical data of PDAC and healthy subjects have been collected.

### 2.2. Blood Samples Collection

Blood samples and plasma have been collected and stored according to the procedure already described in [25]. The Ethical Committee of the University Campus Bio-Medico di Roma approved this study (Prot. 10/12 ComEt CBM).

### 2.3. Preparation of Graphene Oxide Nanoflakes

GO was purchased from Graphenea (San Sebastián, Spain). GO water dispersion at 0.25 mg/mL was subjected to 2 min of pulsed sonication (Vibra cell sonicator VC505, Sonics, and Materials, United Kingdom) to obtain a final well-dispersed solution of GO nanosheets.

### 2.4. Preparation of Graphene Oxide-Protein Corona (GO-PC) Complexes

HP samples derived from healthy and PDAC-affected subjects were diluted 1:10 with ultrapure water. Next, 100 µL of GO solution (see Section 2.3) were incubated with 5% *v*/*v* of HP for 1 h at 37 °C. After 1 h, the solutions were centrifuged at 18,620 RCF for 20 min at 4 °C, then the supernatant was removed, and the pellet was washed in 200 µL of ultrapure water. To fully remove the unbound proteins, the centrifugation was repeated three times. Finally, the last pellets were used for the next PC characterization through sodium dodecyl sulfate polyacrylamide gel electrophoresis (SDS-PAGE).

### 2.5. Size and Zeta-Potential Experiments

For size and zeta-potential measurements, GO was diluted 10 times with ultrapure water. Size and zeta-potential experiments were performed through dynamic light scattering (DLS) and micro-electrophoresis (ME) using a Zetasizer Nano ZS90 equipped with a 5 mW HeNe laser with a wavelength equal to 633 nm and a digital logarithmic correlator (Malvern, United Kingdom). Size and zeta-potential measurements were performed at room temperature. The results are reported as average ± standard deviation of three independent measurements.

### 2.6. Atomic Force Microscopy

To assess GO morphology, atomic force microscopy (AFM) experiments were performed. Of note, 20 µL of GO were deposited on sterile freshly cleaved mica discs, air-dried, and measured with a NanoWizard II atomic force microscope (JPK Instruments AG, Berlin, Germany). The acquisition of AFM images was performed using silicon cantilevers with high-aspect-ratio conical silicon tips (CSC37 Mikro-Masch, Tallinn, Estonia) with an end radius of about 10 nm and a half conical angle of 20°. Data analysis was computed by JPK instrument software. Further details can be found in previous works [26].

### 2.7. 1D SDS-PAGE Experiments

Pellets composed of GO-PC complexes (see Section 2.4) were suspended in 20 μL of Laemmli loading buffer 1×, boiled at 100 °C for 10 min, and centrifuged at 18,620 RCF for 15 min at 4 °C. Finally, 10 μL of supernatants containing the proteins were collected and loaded on a stain-free gradient polyacrylamide gel (4–20% TGX precast gels, Bio-Rad, Hercules, CA, USA) and run at 150 V for about 100 min. Gel images were obtained with a ChemiDocTM imaging system (Bio-Rad, Hercules, CA, USA) and processed by ImageLab Software and custom Matlab (MathWorks, Natick, MA, USA) scripts.

### 2.8. Statistical Data Analysis

Classification of PDAC and healthy subjects was performed by multivariate analysis. Briefly, the outcome from 1D SDS-PAGE experiments, i.e., the integral areas of the profiles within 20–30 kDa and 37–80 kDa, was coupled to the circulating level of plasma proteins belonging to the acute phase protein family. Linear discriminant analysis (LDA) was carried out to evaluate the classification ability of the test in terms of specificity, sensitivity, and global accuracy. The corresponding receiver operating characteristic (ROC) analyses are also provided for evaluating the performance of diagnostic tests, in terms of ROC curves and area under the curve (AUC). Statistical data analysis was performed with Matlab (MathWorks, Natick, MA, USA, Version R2022a) software.

### 2.9. Nanoliquid Chromatography Tandem Mass Spectrometry

GO nanoflakes (0.25 mg/mL) were incubated with HP (5% *v*/*v*) for 1 h at 37 °C. Then, samples were centrifuged three times at 18,620 RCF for 20 min at 4 °C to remove loosely bound proteins. After each time, the pellet was resuspended in ultrapure water. Then, pellets were treated for protein denaturation, digestion, and desalting following a robust protocol that is generally applied to isolate unbound and loosely bound proteins from bio-coronated materials [27]. Finally, the samples were lyophilized using a Speed-Vac device (mod. SC 250 Express; Thermo Savant, Holbrook, NY, USA), reconstituted with 0.1% HCOOH solution, and stored at −80 °C until use. Tryptic peptides were investigated by using a nanoliquid chromatography apparatus (Dionex Ultimate 3000, Sunnyvale, CA, USA) linked to a hybrid mass spectrometer (Thermo Fisher Scientific, Bremen, Germany) with a nanoelectrospray ion source. Xcalibur (v.2.07, Thermo Fisher Scientific) raw data files were submitted to Proteome Discover (1.2 version, Thermo Scientific) for a database search using Mascot (version 2.3.2 Matrix Science). Data were searched against the SwissProt database (v 57.15, 20266 sequences) using the decoy search option of Mascot, and protein quantification was performed using Scaffold software. For each identified protein, the mean value of the normalized spectral countings (NSCs) was normalized to the protein molecular weight to obtain the relative protein abundance (RPA). For each identified protein, the reported RPA is the mean of three independent technical replicates ± standard deviation.

## 3. Results and Discussion

A total of 75 subjects, 50 PDACs, and 25 non-oncological patients, have been studied.


*“Among the oncological series, common PDACs amounted for 88% (44 cases), IPMN derived PDACs have been detected in 5 (10%) cases while mixed IPMN-MCN tumor has been found in one (2%) patient.”*


Their demographic and clinical characteristics are reported in Table 1.

Preliminary experiments were aimed at characterizing the size and zeta-potential distributions of GO nanosheets, as well as their thickness. The results displayed in Appendix A show that GO nanosheets were homogeneous in size, negatively charged, and had a uniform thickness of about 1.2 nm. Next, GO nanosheets were incubated with HP from N = 50 PDAC patients and N = 25 healthy subjects for 1 h at room temperature, leading to the formation of personalized protein coronas. For each of the 75 samples, plasma proteins were isolated from GO nanosheets following consolidated procedures [28] and characterized by 1D SDS-PAGE (all the SDS-PAGE gel images are reported in Appendix A). Figure 1, panel a, shows the average 1D molecular weight (MW) distributions from PDAC patients (orange solid line) and healthy volunteers (black solid line).

Significant variations were detected in the two MW ranges 20–30 kDa and 30–87 kDa. The integral areas of these two regions were therefore used as input parameters for running the NEB test (Figure 1, panel b). A clear separation among PDAC patients (orange points) and healthy volunteers (black points) was observed. An LDA returned a sensitivity of 84% and a specificity of 76%, which is in line with previous findings [25]. Receiver operating characteristic (ROC) analysis is reported in Figure 2.

As the next step of this work, we run the NEB test using sex-disaggregated data. As can be seen in Appendix A, the classification accuracy of the test was not significantly influenced by the sex of the subjects.

Recently, some authors reported an increased risk of PDAC in the presence of alteration of pre-diagnostic serum levels of some proteins belonging to the APPs family. APPs include different proteins whose plasma levels increase in response to the presence of inflammation [29] or other injuries such as the presence of neoplasms [30]. However, being a heterogeneous group of proteins with several biological functions, it is difficult to correlate the alteration of their plasma levels with the presence of a specific tumor such as PDAC. On this basis, we aimed to investigate if, in a setting of a multiplexed strategy, nanotechnology could represent a valid tool to better explain changes in APPs in PDAC patients. We, therefore, developed two versions of the NEB test by coupling the integral area of the region between 20 and 30 kDa with the circulating levels of APPs. Figure 3 shows results for alpha 1 (Figure 3, panel a) and alpha 2 (Figure 3, panel b), respectively. Alpha 1 globulins are a family of serum proteins whose main components are antitrypsin, a protease inhibitor involved in pulmonary alveolar integrity preservation, and serum amyloid A, an inflammatory response modulator also involved in the metabolism of cholesterol. The alpha 2 globulins include proteins such as a C-Protein, haptoglobin, and ceruloplasmin, which play a role in the metabolism of hemoglobin, iron, and copper as well as in the coagulation processes.

By coupling the results of the NEB test with the levels of APPs, interesting results have been highlighted. Particularly, we noticed an increase in the specificity of the test when alpha 1 globulins have been studied. Considering that among the main representatives of the alpha 1 globulins family is the antitrypsin, an inhibitor of trypsin which is normally produced by the pancreas, this result is consistent with what is already reported in the literature about the imbalance of proteinase/antiproteinase and its role in PDAC carcinogenesis [31]. Conversely, when alpha 2 globulins have been coupled, we noticed an increase in the sensitivity of the test. Given that the main representatives of this class of proteins are involved in the metabolism of hemoglobin, and copper and in regulating the formation of thrombi, and that pancreatic cancer is highly cachetizing and prothrombotic [32], we could hypothesize that at the basis of this increase in sensitivity, and therefore, the ability to identify subjects affected by the tumor, there is the alteration of all the aforementioned functions in the presence of the neoplasm. Nonetheless, the association between cachexia and APPs alterations has been already reported in previous studies [33,34].

The results reported in Figure 3 show that coupling the values of the integral area between 20 and 30 kDa with the levels of alpha 2 returned better values in terms of sensitivity and global accuracy and equal 85.4% and 83.3%, respectively. The corresponding ROC curves are reported in Figure 4.

We next asked whether disaggregating the data analysis by sex could provide different outcomes. A version of the test made with sex-disaggregated data provided substantially better classification accuracy for men than for women (Figure 5). The corresponding ROC curves are reported in Figure 6.

This result may have a twofold explanation; first, there could be a bias due to the sample size as females are more represented than males (39 vs. 36 subjects respectively). Second, it is known that differences in concentration of alpha 2 globulins exist between women and men and are mainly related to hormonal influence, as reported in rats [35] and humans [36]. The classification accuracy of the proposed multiplexed test is higher than that of the validated biomarker carbohydrate antigen 19-9 (CA19-9). CA19-9, if elevated, is useful in following patients with known disease [37]. However, the sensitivity and specificity rate of serum CA19-9 alone in diagnosing PDAC have been reported to be low (i.e., about 78.2% and 82.8%, respectively) [38].

To investigate further the diagnostic potential of the test, we performed nano-LC MS/MS experiments. The lists of identified proteins are reported in Appendix A. SDS-PAGE reported in Figure 3 indicated significantly decreased abundances of proteins with MW comprised between 20 and 30 kDa for PDAC patients. Among them, we identified Apolipoprotein A-I (30 kDa) and Apolipoprotein D (21 kDa), Complement factor D (27 kDa), and Complement factor H-related protein 2 (30 kDa). Decreased levels of Apolipoproteins have been already reported in PDAC and suggested as a potential biomarker of this dreadful disease [39,40]. Complement D factor, also called adipsin, is known to be decreased in glucose intolerance and type 2 diabetes [41], as the relation between diabetes and new-onset diabetes and pancreatic cancer is well established [42]. Our proteomic analysis also showed that levels of other proteins are significantly decreased in the presence of pancreatic cancer as tetranectin (22 kDa). This finding is in agreement with the conclusions of a previous investigation that reported a 1.6-fold decrease of tetranectin in the sera of pancreatic cancer patients [43].

On the other hand, SDS-PAGE also showed significantly increased levels of proteins with MW comprised between 37 and 80 kDa for PDAC patients. Among these proteins, nano-LC MS/MS identified a significant increase of different factors involved in the homeostasis of the coagulation process as alpha 2-HS glycoprotein (39 kDa), Fibrinogen gamma chain (51 kDa), Antithrombin-III (52 kDa), Alpha-2-antiplasmin (54 kDa), Fibrinogen beta chain (55 kDa), and Heparin cofactor 2 (57 kDa). These data agree with what was already reported by other authors [43,44], who focused on pancreatic cancer biomarkers discovery, and are consistent with the well-known prothrombotic effect of PDAC [45]. Furthermore, with a molecular weight of 47 kDa and 46 kDa, respectively, our proteomics identified significantly increased levels of Alpha-1-antichymotrypsin and Alpha-1-antitrypsin, which have already been reported as upregulated [46,47] or to be immunochemistry staining markers for pancreatic cancers [48,49].

In conclusion, we underline that this work is not without limitations such as the small size of the population. While the small number of investigated samples in this proof-of-concept work does not allow drawing absolute conclusions, we believe that this promising outcome may represent a starting point for future, more in-depth investigations.

## 4. Conclusions

In the field of pancreatic cancer diagnosis, nanotechnologies are rapidly emerging as useful tools in providing reliable, cheap, and easily reproducible diagnostic tests. In the last few years, researchers have been aiming at identifying new sensitive biomarkers for early-stage PDAC detection. Starting from these considerations, here we developed a multiplexed strategy combining a GO-based variant of the NEB test with the circulating levels of APPs. From one side, a promising approach for PDAC detection was developed. On the other side, we confirmed the relationship between sex-dependent alterations of APPs and the presence of this lethal malignancy. More in general, our result paves the way for the development of multiplexed strategies for early cancer detection and the identification of new potential cancer biomarkers.

## Figures and Tables

**Figure 1 cancers-14-04658-f001:**
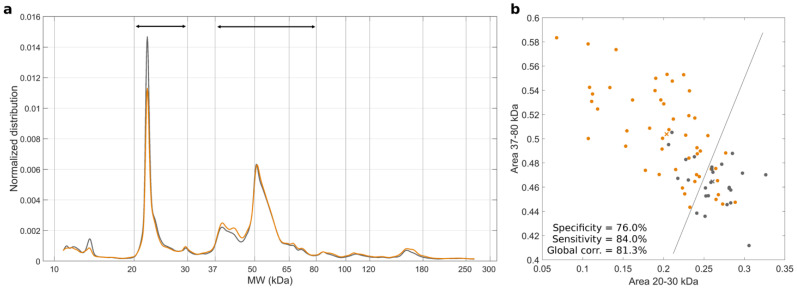
(**a**) Exposing graphene oxide (GO) nanosheets to human plasma (HP) from PDAC patients and healthy volunteers leads to the formation of personalized protein coronas. Protein patterns were isolated from GO and analyzed by 1-dimensional (1D) SDS-PAGE. (**a**) Average molecular weight (MW) distributions of N = 50 PDAC patients (orange solid line) and N = 25 healthy volunteers (black solid line) were obtained by the individual profiles reported in Appendix A. (**b**) Scatter plots of the integral areas show the largest difference between the 1D SDS-PAGE profiles (i.e., those between 20 and 30 KDa, and between 37 and 80 kDa). Each point refers to a single human subject (orange for PDAC and black for healthy participants in the study), while the crosses indicate the centers of the two distributions. The solid black line depicts the output of the linear discriminant analysis for the two distributions. The corresponding receiver operating characteristic (ROC) curve is reported in Figure 2.

**Figure 2 cancers-14-04658-f002:**
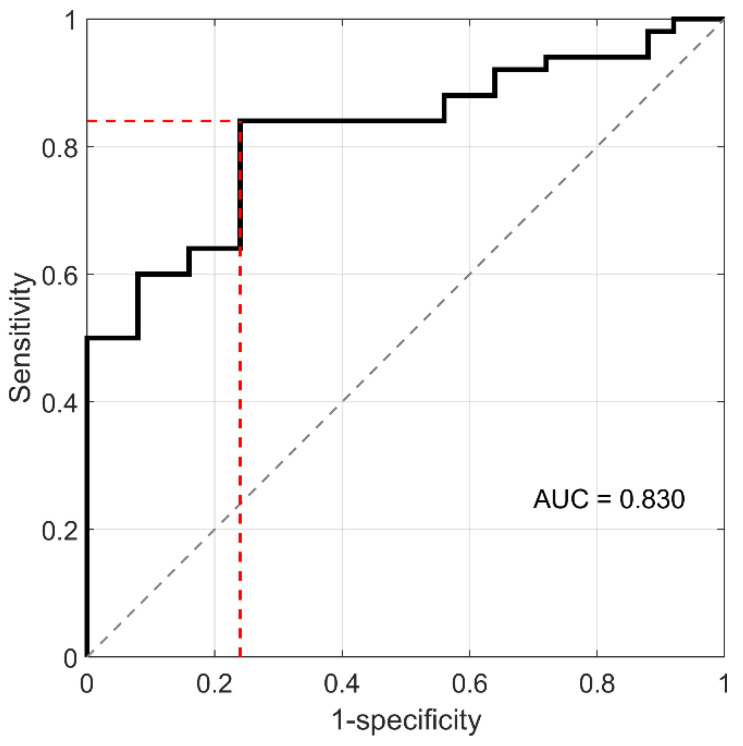
Receiver operating characteristic (ROC) curve obtained by LDA in the two-parameter domain of the 20–30 kDa integral area and 37–80 kDa integral area.

**Figure 3 cancers-14-04658-f003:**
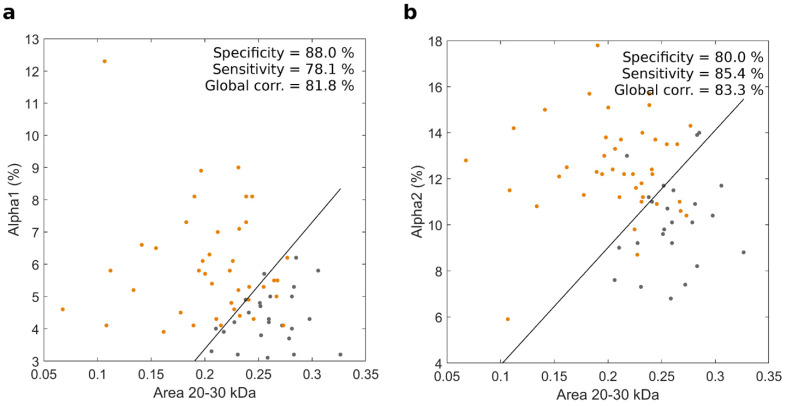
Scatter plots of the integral area showing the largest difference among the 1D SDS-PAGE profiles of PDAC patients and healthy volunteers (i.e., the area between 20 and 30 KDa) coupled to the levels of alpha 1 (panel **a**) and alpha 2 (panel **b**). Each point indicates a single human subject (orange for PDAC and black for healthy participants in the study). The solid black line represents the results of the linear discriminant analysis for the two distributions.

**Figure 4 cancers-14-04658-f004:**
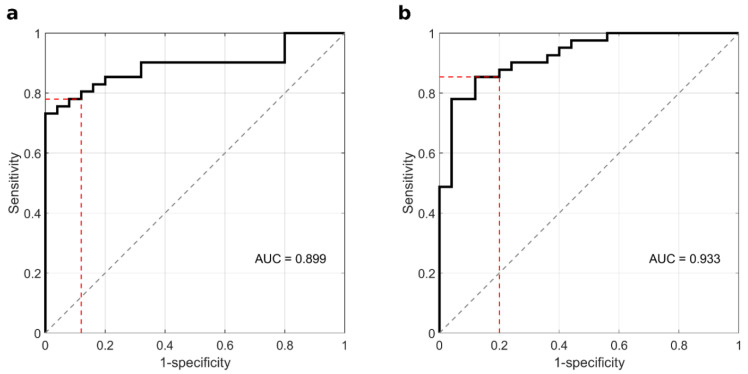
ROC curve obtained by LDA in the following 2-parameter spaces: (**a**) 20–30 kDa integral area and alpha 1 level, and (**b**) 20–30 kDa integral area and alpha 2 level.

**Figure 5 cancers-14-04658-f005:**
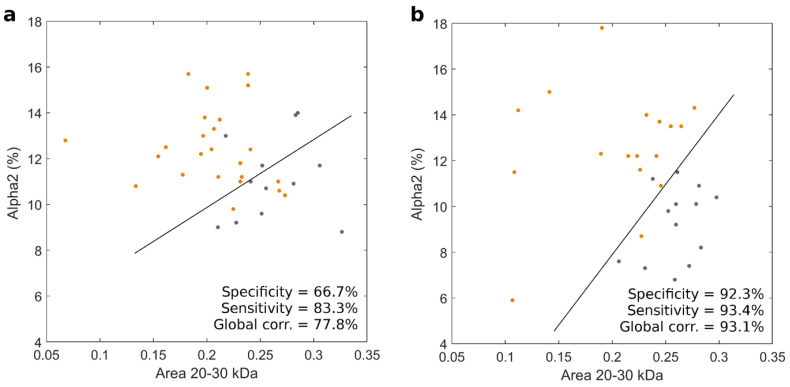
Sex-disaggregated scatter plots of the parameters show the largest difference between the 1D SDS-PAGE profiles for women (panel **a**) and men (panel **b**). Each point indicates a single human subject (orange for PDAC and black for healthy participants in the study). In both panels, the solid black lines describe the result of the linear discriminant analysis for the distributions.

**Figure 6 cancers-14-04658-f006:**
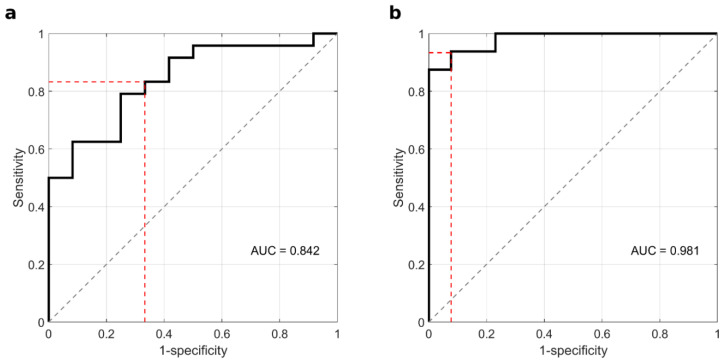
Sex-disaggregated ROC curves for (**a**) women and (**b**) men, obtained by LDA in the 2-parameter space of the 20–30 kDa integral area and the alpha 2 level.

**Table 1 cancers-14-04658-t001:** Demographic and clinic characteristics of the control group and the pancreatic ductal adenocarcinoma (PDAC) group.

Characteristic	Controls (*n* = 25)	PDAC (*n* = 50)
Age, median (IQR), y	55 (40–64)	71 (64.5–76.5)
Sex, No. (%)		
Male	13 (52%)	23 (46%)
Female	12 (48%)	27 (54%)
Pathologies		
Cholelithiasis	13	NA
Groin hernia	3	NA
Umbilical hernia	1	NA
Incisional hernia	2	NA
Hiatal hernia	1	NA
Colonic diverticular disease	3	NA
Muco-hemorroidal prolapse	1	NA
Pilonidalis sinus	1	NA
TNM stage		
I	NA	12
II	NA	15
III	NA	15
IV	NA	8

## Data Availability

The data presented in this study are available upon request from the corresponding author and in respect of local privacy law.

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
