# Peer review of "Multiplexed Detection of Pancreatic Cancer by Combining a Nanoparticle-Enabled Blood Test and Plasma Levels of Acute-Phase Proteins"

_cancers, 2022, doi:10.3390/cancers14194658_

Round 1
Reviewer 1 Report
In the manuscript entitled "Multiplexed detection of pancreatic cancer by combining nanoparticle-enabled blood test and plasma levels of acute-phase proteins", the authors presented their data of distinguishing between pancreatic ductal adenocarcinoma plasma and control plasma, using the incubation with graphene oxide nanomaterial followed by SDS PAGE. This assumption is very interesting, but the current data are not solid enough to support the conclusions. The following concens should be addressed before further processing of the manuscript.
- Nanoparticle and nanoflakes are both used in the manuscript. The nanomaterial should be characterized further, at least using TEM or SEM, to clarify the morphology of the nanomaterial.
- Based on the data presented, the diagnostic potential of this method needs to be further improved.
- If a band with diagnostic potential can be found, the proteins in this band should be further identified.
Author Response
In the manuscript entitled "Multiplexed detection of pancreatic cancer by combining nanoparticle-enabled blood test and plasma levels of acute-phase proteins", the authors presented their data of distinguishing between pancreatic ductal adenocarcinoma plasma and control plasma, using the incubation with graphene oxide nanomaterial followed by SDS PAGE. This assumption is very interesting, but the current data are not solid enough to support the conclusions. The following concens should be addressed before further processing of the manuscript.
Q1. Nanoparticle and nanoflakes are both used in the manuscript. The nanomaterial should be characterized further, at least using TEM or SEM, to clarify the morphology of the nanomaterial.
Reply: We thank the Reviewer for Her/His comment. We added atomic force microscopy (AFM) analysis in the manuscript to characterize further the employed nanomaterial. Results are reported in the Supplementary Materials.
Q2. Based on the data presented, the diagnostic potential of this method needs to be further improved.
Reply: We thank the Reviewer for Her/His comment. To address this concern, receiving operating characteristic (ROC) analyses have been added in the work (all reported in the Supplementary Materials), as well as the identification of proteins populating the main bands of the electrophoretic outcomes. This last aspect is also discussed as a reply to Q3.
Q3. If a band with diagnostic potential can be found, the proteins in this band should be further identified.
Reply: We thank the Reviewer for Her/His comment. In previous works, we focused on the analysis of the protein content in the corona of GO surface. This has great relevance for the diagnostic potential of protein-corona based tests. As an instance, NanoLiquid Chromatography MS/MS experiments revealed that the most prominent protein in the 20-30 kDa range is Apolipoprotein A1 (APOA1) (Di Santo, Riccardo, et al. "Personalized graphene oxide-protein corona in the human plasma of pancreatic cancer patients." Frontiers in bioengineering and biotechnology 8. 2020), which has a high affinity to GO (Castagnola, Valentina, et al. "Biological recognition of graphene nanoflakes." Nature communications 2018) and, interestingly, is recognized as a potential biomarker for PDAC (Lin, Chao, et al. "ITRAQ-based quantitative proteomics reveals apolipoprotein AI and transferrin as potential serum markers in CA19-9 negative pancreatic ductal adenocarcinoma." Medicine 2016.). This outcome has been confirmed by further experiments, in which other protein contributions were identified, including immunoglobulin lambda like polypeptide 5 (IGLL5). Serum albumin (ALBU), immunoglobulin heavy constant mu (IGHM), and Alpha-1-B glycoprotein (A1BG) were the most abundant proteins in the 37-80 kDa range (Digiacomo, Luca, et al. "Magnetic Levitation Patterns of Microfluidic-Generated Nanoparticle–Protein Complexes." Nanomaterials. 2022). Noteworthy, A1BG has been found to be overexpressed in the pancreatic juice from PDAC patients (Tian, Mei, et al. "Proteomic analysis identifies MMP-9, DJ-1 and A1BG as overexpressed proteins in pancreatic juice from pancreatic ductal adenocarcinoma patients." BMC cancer 2008).
In this work, the outcome of NanoLiquid Chromatography MS/MS experiments are reported in Tab. S2 and discussed at the end of the Results and discussion section.
Reviewer 2 Report
Dear editors:
It is a great honor and pleasure for me to be invited as a reviewer for the manuscript entitled “Multiplexed detection of pancreatic cancer by combining nanoparticle- enabled blood test and plasma levels of acute-phase proteins?”. Damiano Caputo and the co-authors have evaluated the association between serum levels of acute-phase proteins (APPs) and pancreatic ductal adenocarcinoma (PDAC). I have a number of major problems concerning this study:
1. Pancreatic cancer types can be divided into two large categories: exocrine PDCA and neuroendocrine pancreatic cancer. Each category has diverse cancer types that neuroendocrine pancreatic cancer should not be excluded.
2. According to the main results of the article, the causation is not deterministic. Their data showed the “correlation” between the circulating levels of alpha 2 globulins and pancreatic cancer events (Fig. 2 and Fig. 3). However, they concluded that alpha 2 globulins had better predictive power for men than for women in Fig 3. Meanwhile, they admit their bias as follows: (1) the sample size: females are more represented than males; (2) the differences in concentration of alpha2 globulins exist between women and men and are mainly related to hormonal influence. Moreover, the sample size is too small that data should be interpreted with caution. In my opinion, the sample size of the controls should be expanded. While figuring out a diagnostic test's accuracy and a cut-off point of the test for a disease, receiver operating characteristic curve (ROC) analysis must be provided for clinical application. Despite the high global correlation between alpha2 globulins and PDCA, current results lacking in generalizability and reliability (reproducibility) could not be a gold standard for PDCA diagnosis.
3. In light of the nature of alpha1-antitrypsin, information of pancreatitis, pulmonary and hepatic injury should be provided.
4. The validated biomarker, CA-199, should be compared with the predictive power of the current biomarker alpha 2 globulins.
5. The term “predict” should be avoided and rephrased: Line 36, 92, 205, 241, 249.
6. Line 73: The abbreviation “GO” is used. GO should be used in Line 113.
7. The purpose and rationale is not clear enough in the introduction, e.g., the sentence in Line 90 should be placed in other section; the sentence in Line 91 should be placed in discussion section.
8. Line 100: the definition of benign surgical diseases (e.g., cholelithiasis abdominal hernia, hemorrhoids, etc.) should be described in detail. Why did they include 8 diseases and exclude other diseases in Fig. 1?
9. In the method: blood urea nitrogen did not always reflect renal function, but albuminuria did (CKD staging in KDIGO guideline).
10. Line 103: Does the term “blood disease” include anemia? Does the term “tumors” mean all kinds of benign and malignant tumors? The sample size is only 75.
11. Line 73: The abbreviation “HP” is used. HP could be used in Line 118.
12. Line 73: The abbreviation “MWNSC” is not necessary.
Author Response
It is a great honor and pleasure for me to be invited as a reviewer for the manuscript entitled “Multiplexed detection of pancreatic cancer by combiningnanoparticle-enabled blood test and plasma levels of acute-phase proteins”. Damiano Caputo and the co-authors have evaluated the association between serum levels of acute-phase proteins (APPs) and pancreatic ductal adenocarcinoma (PDAC). I have a number of major problems concerning this study:
Q1. Pancreatic cancer types can be divided into two large categories: exocrine PDCA and neuroendocrine pancreatic cancer. Each category has diverse cancer types that neuroendocrine pancreatic cancer should not be excluded.
Reply: We thank the Reviewer for Her/His comment that is totally right. According to this suggestion we reported data regarding the number of PDAC deriving from IPMN and those regarding malignant cystic neoplasm (MCN). The following sentence has been added and highlighted in the results section: “Among the oncological series, common PDAC amounted for 88% (44 cases), IPMN derived PDAC have been detected in 5 (10%) cases while mixed IPMN-MCN tumour has been found in one (2%) patient.”
Q2. According to the main results of the article, the causation is not deterministic. Their data showed the “correlation” between the circulating levels of alpha 2 globulins and pancreatic cancer events (Fig. 2 and Fig. 3). However, they concluded that alpha 2 globulins had better predictive power for men than for women in Fig 3. Meanwhile, they admit their bias as follows: (1) the sample size: females are more represented than males; (2) the differences in concentration of alpha2 globulins exist between women and men and are mainly related to hormonal influence. Moreover, the sample size is too small that data should be interpreted with caution. In my opinion, the sample size of the controls should be expanded. While figuring out a diagnostic test's accuracy and a cut-off point of the test for a disease, receiver operating characteristic curve (ROC) analysis must be provided for clinical application. Despite the high global correlation between alpha2 globulins and PDCA, current results lacking in generalizability and reliability (reproducibility) could not be a gold standard for PDCA diagnosis.
Reply: We thank the Reviewer for Her/His comment. We agree that the data must be interpreted with caution and that absolute conclusions should not be drawn. We also believe that, given the occurrence of PDAC, using n=50 samples is not so bad. Anyway, to address the point raised by the reviewer the following paragraph has been added to the revised version of the manuscript. It reads: “In conclusion, we underline that this work is not without limitations such as the small size of the population. While the small number of investigated samples in this proof-of-concept work does not allow drawing absolute conclusions, we believe that this promising outcome may represent a starting point for future more in-depth investigations.” Lastly, to meet the reviewer’s comment, we provided ROC analyses (Fig. S5, S7, and S8 in the Supplementary Materials). Briefly, results from ROC analyses confirmed the main trends of the NEB test’s parameters.
Q3. In light of the nature of alpha1-antitrypsin, information of pancreatitis, pulmonary and hepatic injury should be provided.
Reply: We thank the Reviewer for Her/His comment that is totally right. According to this suggestion we improved the Methods section reporting among the inclusion criteria the absence of pancreatitis, lung and hepatic diseases. The Materials and Methods section has been modified as follow: “ The inclusion criteria for both groups were…. absence of uncontrolled infections, pancreatitis, lung and hepatic coexistent diseases...” Thanks again to the Reviewer for this important comment.
Q4. The validated biomarker, CA-199, should be compared with the predictive power of the current biomarker alpha 2 globulins.
Reply: We thank the Reviewer for Her/His comment. As suggested by the Reviewer in Q5, we replaced the term “predictive power” to “classification accuracy”, as it is more appropriate for the performed analysis. The multiplexed test based on alpha2 globulin and electrophoretic outcome exhibited 83.3% classification accuracy, that reached 93.1% in sex-disaggregated analysis. These values can be compared to the validated biomarker carbohydrate antigen 19-9 (CA19-9). CA19-9 if elevated, it is useful in following patients with known disease (Ryan, David P., Theodore S. Hong, and Nabeel Bardeesy. "Pancreatic adenocarcinoma." New England Journal of Medicine 371.11 (2014): 1039-1049.). However, sensitivity and specificity rate of serum CA19-9 alone in diagnosing PDAC have been reported to be low (i.e. about 78.2% and 82.8%, respectively) (Yang, Chi-Ying, et al. "Accuracy of simultaneous measurement of serum biomarkers: Carbohydrate antigen 19-9, pancreatic elastase-1, amylase, and lipase for diagnosing pancreatic ductal adenocarcinoma." Journal of the Formosan Medical Association (2022). Furthermore, it is known that the efficacy of CA 19-9 in predicting either pancreatic cancer or other cancers in the asymptomatic population is low (Chang, Chi-Yang, et al. "Low efficacy of serum levels of CA 19-9 in prediction of malignant diseases in asymptomatic population in Taiwan." Hepato-gastroenterology. 2006). Taken together, these considerations suggest that the multiplexed test have better diagnostic potential than the validated biomarker CA 19.9. These aspects have been added in the Results and discussion section.
Q5. The term “predict” should be avoided and rephrased: Line 36, 92, 205, 241, 249.
Reply: We thank the Reviewer for Her/His comment. The term “predictive power” has been replaced with “classification accuracy” in all the indicated lines.
Q6. Line 73: The abbreviation “GO” is used. GO should be used in Line 113.
Reply: We thank the Reviewer for Her/His comment. The extended form “Graphene Oxide” has been replaced with the abbreviation “GO”.
Q7. The purpose and rationale is not clear enough in the introduction, e.g., the sentence in Line 90 should be placed in other section; the sentence in Line 91 should be placed in discussion section.
Reply: We thank the Reviewer for Her/His comment. The sentence in Line 90 has been rephrased, and the sentence in Line 91 has been placed in the discussion section.
Q8. Line 100: the definition of benign surgical diseases (e.g., cholelithiasis abdominal hernia, hemorrhoids, etc.) should be described in detail. Why did they include 8 diseases and exclude other diseases in Fig. 1?
Reply: We thank the Reviewer for Her/His comment. All the pathologies that by their nature could influence the concentration of acute phase proteins have been excluded from the analysis. For this reason, other acute benign pathologies usually cured in our Department of Surgery, have been excluded. Thanks to the reviewer's comment, we specified this concept in the Materials and Methods section adding the following sentence: “Patients affected by acute diseases, that by their nature could influence the APPs concentration (e.g., peritonitis, bowel obstruction, etc.), have been excluded from the analysis such as patients underwent urgent surgical procedures were.”
Q9. In the method: blood urea nitrogen did not always reflect renal function, but albuminuria did (CKD staging in KDIGO guideline).
Reply: We thank the Reviewer for Her/His comment and totally agree with Her/Him. For this reason, as stated in the Methods section we also considered creatinine levels. In our series, albuminuria assay has not routinely performed due to the absence of history of chronic renal failure of the patients. Therefore, unfortunatlely, these data are non available. However, according to the guidelines, we calculated the GFR values and did not find patients in G3 categories. The great majority of the subjects in both groups were G1. Furthermore, even the few patients who presented in category G2 did not have changes for more than 3 months. We thank the Reviewer for this suggestion that allowed to improve our manuscript. We reported the following sentences in the Methods sections: “The inclusion criteria... normal renal function (creatinine < 1.5 mg/dL, blood urea nitrogen < 1.5 times the upper limit) or mildly decreased renal function (< G3 on the basis of Glomerular Filtration Rate score proposed by KDIGO guidelines) for less than 3 months; no previous personal medical…”.
(da confermare)
Q10. Line 103: Does the term “blood disease” include anemia? Does the term “tumors” mean all kinds of benign and malignant tumors? The sample size is only 75.
Reply: We thank the Reviewer for Her/His comment. Anemia was an exclusion criteria only if chronic. We meant malignant tumors. We apologies with the Reviewer for not being clear and modified the Methods section as follows: “…no previous personal medical history of malignant tumors and chronic blood disease;…”
Q11. Line 73: The abbreviation “HP” is used. HP could be used in Line 118.
Reply: We thank the Reviewer for Her/His comment. The extended form “Human Plasma” has been replaced with the abbreviation “HP”.
Q12. Line 73: The abbreviation “MWNSC” is not necessary.
Reply: We thank the Reviewer for Her/His comment. The abbreviation has been removed.
Reviewer 3 Report
The author provides the method for pancreatic cancer (PDAC) detection using the combination of graphene oxide (GO) nano sheets made the corona from blood plasma (acute-phase proteins) of 25 patients and 50 PDAC patients. Identified the biomarkers using the SDS-PAGE AND nano-LC MS/MS.
major comments:
1. What are the other methods or commercial clinical methods (similar to blood Methods or multiplexed methods) for the early detection of PDAC and discuss those methods in the manuscript in the discussion.
2. author emphasized the early detection of PDAC. how this multiplex method is cost and time effective for diagnosis?
Author Response
graphene oxide (GO) nano sheets made the corona from blood plasma (acute-phase proteins) of 25 patients and 50 PDAC patients. Identified the biomarkers using the SDS-PAGE AND nano-LC MS/MS.
Q1. What are the other methods or commercial clinical methods (similar to blood Methods or multiplexed methods) for the early detection of PDAC and discuss those methods in the manuscript in the discussion.
Reply: We thank the reviewer for her/his suggestion. Many other methods have been developed over the years for early PDAC detection. In the last few years, researchers have been aiming at identifying new sensitive biomarkers for early-stage PDAC detection. The most promising results were obtained by advances in molecular technologies such as gene sequencing, transcriptomic, glycomics and proteomics. Although the number of novel biomarkers (e.g., mi-RNAs, , circulating tumor DNA, circulating tumor cells, and exosomes) has increased over the last years, there is still no evidence of their efficacy since most of them lack in sensitivity and specificity. Of note, proteomic techniques are among the most powerful tools for potential biomarker identification by mass spectrometry analysis. However, the high-cost production and the laborious and time-consuming procedures have spurred researchers to look at more accessible resources. Among these, PC-based diagnostic tools are emerging as the most promising approaches for early cancer detection, since reserve several benefits such as low cost of investigation, high sensitivity and specificity, low intra-individual variability. These aspects have been added and discussed in the introduction of the manuscript.
Q2. author emphasized the early detection of PDAC. how this multiplex method is cost and time effective for diagnosis?
Reply: The presented multiplexed strategy allows a rapid, low-cost and efficient discrimination between healthy and PDAC-affected subjects by combining the relative clinical levels of circulating plasma proteins and the NEB test outcomes. In fact, the electrophoretic approach has been largely optimized over the years until an adequate standardization that guarantees high efficiency, accuracy, and rapidity. The use of the SDS-PAGE for the PC characterization assures low-cost investigation and immediate results. Moreover, the analysis of the protein patterns has also been automated by the use of dedicated software that allows rapid and highly reproducible outcomes even when a large sample size is requested. Compared to proteomic techniques that are employed for the identification of PDAC’s biomarker, the presented one represents a faster and cheaper approach, specifically designed for diagnostic purposes. These aspects have been added in the Introduction section.
Round 2
Reviewer 1 Report
The revised version of the manuscript has been improved, and it could be processed further.
Author Response
The reviewer did not ask for further changes.Reviewer 2 Report
Dear editors and authors,
The authors work hard to improve their scientific merits. I think all of the ROC analysis could be included in the article, instead of supplemental data.
Thank you so much.
Author Response
The ROC analysis has been included in the article.